# Longitudinal Assessment of BNT162b2- and mRNA-1273-Induced Anti-SARS-CoV-2 Spike IgG Levels and Avidity Following Three Doses of Vaccination

**DOI:** 10.3390/vaccines12050516

**Published:** 2024-05-09

**Authors:** Jimmie L. Bullock, Thomas E. Hickey, Troy J. Kemp, Jordan Metz, Sarah Loftus, Katarzyna Haynesworth, Nicholas Castro, Brian T. Luke, Douglas R. Lowy, Ligia A. Pinto

**Affiliations:** 1Vaccine, Immunity and Cancer Directorate, Frederick National Laboratory for Cancer Research, Frederick, MD 21702, USA; jimmie.bullock@nih.gov (J.L.B.J.); thomas.hickey@nih.gov (T.E.H.); kemptj@mail.nih.gov (T.J.K.); jordan.metz@nih.gov (J.M.); sarah.loftus@nih.gov (S.L.); katrina.haynesworth@nih.gov (K.H.);; 2Bioinformatics and Computational Science Directorate, Frederick National Laboratory for Cancer Research, Frederick, MD 21702, USA; lukebria@mail.nih.gov; 3Laboratory of Cellular Oncology, National Cancer Institute, National Institutes of Health, Bethesda, MD 20892, USA; lowyd@mail.nih.gov

**Keywords:** immunoglobulin, SARS-CoV-2, vaccination, avidity, longitudinal, ELISA

## Abstract

SARS-CoV-2 vaccination-induced protection against infection is likely to be affected by functional antibody features. To understand the kinetics of antibody responses in healthy individuals after primary series and third vaccine doses, sera from the recipients of the two licensed SARS-CoV-2 mRNA vaccines were assessed for circulating anti-SARS-CoV-2 spike IgG levels and avidity for up to 6 months post-primary series and 9 months after the third dose. Following primary series vaccination, anti-SARS-CoV-2 spike IgG levels declined from months 1 to 6, while avidity increased through month 6, irrespective of the vaccine received. The third dose of either vaccine increased anti-SARS-CoV-2 spike IgG levels and avidity and appeared to enhance antibody level persistence—generating a slower rate of decline in the 3 months following the third dose compared to the decline seen after the primary series alone. The third dose of both vaccines induced significant avidity increases 1 month after vaccination compared to the avidity response 6 months post-primary series vaccination (*p* ≤ 0.001). A significant difference in avidity responses between the two vaccines was observed 6 months post-third dose, where the BNT162b2 recipients had higher antibody avidity levels compared to the mRNA-1273 recipients (*p* = 0.020).

## 1. Introduction

Human infections with severe acute respiratory syndrome coronavirus 2 (SARS-CoV-2) rapidly escalated to the global pandemic of coronavirus disease 2019 (COVID-19) in March 2020 [1,2,3]. Two messenger RNA (mRNA) vaccines, BNT162b2 (Pfizer-BioNTech) and mRNA-1273 (Moderna), containing SARS-CoV-2 spike glycoprotein mRNA were rapidly developed to address public health needs and have been used extensively in the United States [4,5,6]. Both vaccines showed excellent efficacy via the elicitation of neutralizing antibodies to the original virus strain up to 6 months after vaccination [7,8]; however, many studies have confirmed low antibody titers at later timepoints [9,10,11,12,13]. These observations led to booster recommendations in the United States [9,10,11,12,13,14,15,16]. However, the circulating antibody level is unlikely to be the only important measurement to assess immunity; the quality of the antibody, or avidity, most certainly would also hold importance in understanding vaccine immunogenicity [17,18,19].

Vaccine-induced protection through adaptive immunity may be dependent not only on circulating antibody levels but also on functional antibody features such as antibody affinity and avidity [20]. Affinity can be defined as the strength of a single noncovalent interaction between an antibody and an epitope [21]. Avidity, meanwhile, is described as the net strength of binding between a specific target epitope and all antibodies present in the sera [17]. Essentially, avidity is the accumulated binding strength of multiple antibody affinities [21]. In a recent study aiming to identify therapeutic antibodies against SARS-CoV-2 variants, higher-avidity antibodies were more likely to neutralize the later emerging Omicron variants of the SARS-CoV-2 virus [20]. In previous studies, we demonstrated that, while antibody levels had dropped by 4 months in healthy individuals after SARS-CoV-2 mRNA vaccination, avidity increased by month 4 [13]. In this study, an expanded cohort was used to evaluate the kinetics of SARS-CoV-2 spike IgG avidity and antibody responses in healthy individuals up to 6 months after BNT162b2 or mRNA-1273 primary series vaccination and up to 9 months after a third vaccine dose. The intent of this study is to contribute to the growing knowledge base on the quality and persistence of adaptive immunity elicited by mRNA vaccination over time and to evaluate the impact of a third dose of vaccination on long-term avidity development.

## 2. Materials and Methods

### 2.1. Samples

#### 2.1.1. SARS-CoV-2 mRNA Vaccination 

Sera were collected from 109 healthy donors by Occupational Health Services at the National Institutes of Health (NIH) National Cancer Institute (NCI) at Fort Detrick, MD, under the Research Donor Protocol (RDP). Participants were healthy NCI-Frederick employees and other NIH staff that donated blood samples for in vitro research at the NCI-Frederick laboratories. The protocol is listed under NIH protocol number OH99CN046 and NCT number NCT00339911. In this study, all donors had received 2 doses of either BNT162b2 (30 µg, n = 57) or mRNA-1273 (100 µg, n = 52). Samples were obtained at months 1, 4, and 6 following the 2 originally recommended doses (primary series). The administration of the third dose of vaccine occurred 6 to 17 months after the completion of the primary series, resulting in 68 vaccinated participants receiving a third dose of the same vaccine as their primary series (BNT162b2 30 µg, n = 36; mRNA-1273 50 µg, n = 32), and they were subsequently followed for up to 9 months. Donors receiving a heterologous third dose (a dose of vaccine of a different type than their primary series) were not included in this study to simplify the comparison of the effects of the 2 vaccines. Sera were collected from donors at months 1, 3, 6, and 9 after the third dose. Sex, age, and detailed information at collection time points are listed in Table 1 and Table 2. 

#### 2.1.2. Sample Preparation

Blood samples collected in serum tubes were processed and frozen the same day as collection. Serum samples were maintained at −80 °C until tested, then thawed on wet ice, aliquoted, and heat-inactivated at 56 °C for 30 to 60 min prior to testing.

### 2.2. Laboratory Analyses

#### 2.2.1. Enzyme-Linked Immunosorbent Assay (ELISA)

ELISA assays used to quantify human serum IgG antibodies to the SARS-CoV-2 spike protein were performed at room temperature (RT) as previously described [13]. Maxisorp 96-well plates (Thermo-Scientific Cat# 439454, Waltham, MA, USA) were coated with recombinant SARS-CoV-2 soluble spike protein (SARS-CoV-2 S-2P [15-1213]-T4f-His6) sourced from the Protein Expression Laboratory (PEL) at Frederick National Laboratory for Cancer Research (FNLCR) (0.3 µg/mL in phosphate-buffered saline [PBS]). After coating for a minimum of 24 h at 4 °C, assay plates were washed with a PBS–Tween buffer and blocked with PBS–Tween 0.2% blocking buffer (PBS, 0.2% Tween 20, pH 7.4, and 4% skim milk [BD, Cat# 232100] for 90 min to reduce nonspecific binding. After the removal of the blocking buffer, 10 µL serum samples at 1:150 or greater dilution in blocking buffer were three-fold serial diluted, added to wells, and incubated for 1 h at 20 °C. Following incubation, the plates were washed and incubated for 60 min with 100 µL of a 0.067 µg/mL solution of goat anti-human IgG HRP-conjugate (Seracare, Cat# 5220-0390, Milford, MA, USA) in blocking buffer. The plates were washed following incubation and developed with tetramethylbenzidine (TMB) 2-component substrate (Seracare, Cat# 5120-0049, 5120-0038) for 25 min. Finally, the reaction in the plate was stopped with 0.36 N sulfuric acid and read at 450 nm and 620 nm on a SpectraMax plate reader (Molecular Devices, San Jose, CA, USA). Data analyses were performed using SoftMax Pro GxP 7.0.3. Reportable values for anti-SARS-CoV-2 spike IgG quantitative ELISAs are binding antibody units per milliliter (BAU/mL), based on a standard calibrated to the WHO International Standard [22]. 

#### 2.2.2. Avidity Enzyme-Linked Immunosorbent Assay (Chaotrope ELISA)

The avidity ELISA (chaotrope ELISA) is based on the ELISA test for anti-SARS-CoV-2 spike IgG but includes an additional step where the analyte (antibody) is exposed to a chaotropic agent that effectively breaks and elutes off weakly bound antibodies, and it is designated as a “bind and break” ELISA. Urea was used as the chaotropic agent in this study due to its experimental range and minimal impact on the spike protein coated on the assay plate [13]. Chaotrope ELISA assays to assess serum avidity were performed with sample dilutions with optical densities (ODs) between 0.5 and 1.3 OD units at 450 nM; 1.0 was the target OD. Extensive assay development with multiple SARS-CoV-2 serum samples demonstrated highly reproducible (Coefficient of Variation [CV] < 10%) measurements in this OD range. Each assay plate tested 5 serum samples in duplicate. The assay was performed following the same procedures summarized above with additional urea incubation after sample incubation in the plate, which was performed at 22 °C (room temperature [RT]). The plates were washed and incubated with dilutions of urea ranging from 0 to 7 molar concentrations (M) for 15 min at 22 °C, followed by 4 washes in PBS–Tween. Serum avidity assessments are reported as Avidity Indices (AI80), the molar concentration (M) of chaotrope required to reduce the optical density of the sample to 80% that of untreated wells. High- and low-positive system suitability controls, developed from well-characterized serum samples, were added to each plate and utilized to assess the fitness of the assay components and equipment. In the event that either system suitability control failed, all samples in the assay plate were repeated. The failure of controls was characterized by an OD outside of acceptance criteria of 0.5 to 1.3 OD units at 450 nm and/or an AI80 value outside of the established ranges of 2.3–3.2 M and 4.0–4.9 M for low- and high-positive controls, respectively. 

### 2.3. Assay Data Analyses

ELISA data were analyzed with SoftMax Pro^®^, 7.0.3 GxP. AI80 values were calculated by the program and exported to an Excel^®^ database. Analyses of database entries were conducted with Microsoft^®^ Excel software 2021 or Graphpad© Prism 9. 

Serum antibody levels and avidity are expressed as geometric mean concentrations (GMTs) and geometric mean avidity levels (GMAs), respectively, with corresponding 95% confidence intervals (CIs). The avidity assay has an upper limit of 7.0 M, so any sample with a potentially larger value was recorded as 7.0 M. The mean and variance of these values cannot be determined, so a nonparametric test was used. A Mann–Whitney U test (Wilcoxon Rank Sum test) was used when comparing ELISA or avidity values between vaccines or between sexes at birth for each manufacturer. A Wilcoxon signed-rank test was used when ELISA or avidity values were compared between different time points (the analysis is within each sample). *p* < 0.05 was considered significant. 

## 3. Results

### 3.1. Sample Demographic Characteristics

The anti-SARS-CoV-2 spike IgG antibody and antibody avidity levels were studied in the sera of 109 healthy recipients of the BNT162b2 (n = 57) or mRNA-1273 vaccine (n = 52). The sex-at-birth and ages of the participants are listed in Table 1. The distribution of females and males in each group is similar, with females representing 56% and 54% of the BNT162b2 and mRNA-1273 cohorts, respectively. 

Detailed information on the collection time is shown in Table 2A. In total, 36 out of the 57 BNT162b2 participants and 32 out of the 52 mRNA-1273 participants received homologous third doses and continued to be followed. The availability of the third dose limited our ability to follow the longevity of immune responses to the primary vaccine series after 6 months, as a new sample collection schedule started once a third dose was administered. Consequently, antibody measurements more than 6 months after the primary series vaccination are not reported. The serum collection schedules are listed in Table 2.

### 3.2. Antibody Responses to SARS-CoV-2 Spike Protein Following Primary Vaccination with BNT162b2 or mRNA-1273

A standardized enzyme-linked immunosorbent assay (ELISA) was utilized to determine serum IgG antibody levels to recombinant SARS-CoV-2 spike protein in binding antibody units per mL (BAU/mL). Consistent with our previous reports at 1 and 4 months [13], both primary series vaccination groups exhibited a significant waning of IgG responses to the SARS-CoV-2 spike after the first month. No significant correlations were observed between sex or age with IgG levels for either vaccine. In the BNT162b2 recipients, the geometric mean levels of anti-SARS-CoV-2 spike IgG antibody fell from 3117 BAU/mL (95% CI, 2604–3730 BAU/mL) to 764 BAU/mL (95% CI, 619–941 BAU/mL), a decline of 76% (Figure 1B,C) by month 4. Likewise, in the mRNA-1273 recipients, the levels declined by 71% from 1 month to 4 months: the geometric mean, 3692 BAU/mL (95% CI, 2889–4717 BAU/mL), fell to 1088 BAU/mL (95% CI, 879–1347 BAU/mL) (Figure 1A,C). This antibody waning continued into the 6-month timepoint; the levels of anti-SARS-CoV-2 spike IgG fell from 1 to 6 months by 86% in the BNT162b2 vaccinees (*p* ≤ 0.001) and 84% in the mRNA-1273 recipients (*p* ≤ 0.001), respectively.

### 3.3. Effects of a Homologous Third Dose on Circulating Anti-SARS-CoV-2 Spike IgG Levels

The effects of a third dose of either BNT162b2 or mRNA-1273 were evaluated in the recipients’ sera by quantitating anti-SARS-CoV-2 spike IgG levels 1, 3, 6, and 9 months after a third homologous dose of the respective vaccine and comparing those findings to the levels detected in the matched samples obtained from the donors that received only the primary vaccination series. No significant correlations were observed between sex or age and the anti-SARS-CoV-2 spike IgG levels for a third dose of either vaccine when comparing at least 10 recipients. A third dose of either vaccine produced statistically significant increases in circulating IgG levels for SARS-CoV-2 spike protein. The BNT162b2 recipients’ geometric mean anti-SARS-CoV-2 spike IgG levels rose from 452 BAU/mL (95% CI, 371–550 BAU/mL) 6 months after the primary vaccination series to 8562 BAU/mL (95% CI, 6241–11747 BAU/mL) one month after dose 3, *p* ≤ 0.001, representing an 18.9-fold increase (Figure 1B,C). Likewise, the mRNA-1273 recipients’ geometric mean anti-SARS-CoV-2 spike IgG levels were 599 BAU/mL (95% CI, 465–772 BAU/mL) 6 months after the primary vaccination series, and these rose to 9369 BAU/mL (95% CI, 6781–12944 BAU/mL) 1 month after dose 3, *p* ≤ 0.001, representing a 15.6-fold increase (Figure 1A,C). Additionally, vaccine-induced antibody levels continued to wane after the third dose, but at a slower rate than what was detected following the primary series vaccination. The observed decline in antibody levels from 1 to 3 months after a third dose of BNT162b2 was 30%, where the geometric mean anti-SARS-CoV-2 spike IgG levels fell from 8562 BAU/mL (95% CI, 6241–11747 BAU/mL) to 6029 BAU/mL (95% CI, 4342–8370). Similarly, the decline in anti-SARS-CoV-2 spike IgG in the recipients of a third mRNA-1273 dose was 37%; the geometric mean levels fell from 9369 BAU/mL (95% CI, 6781–12944 BAU/mL) to 5946 BAU/mL (95% CI, 4520–7823 BAU/mL) (Figure 1A–C). 

### 3.4. Anti-SARS-CoV-2 Spike IgG Avidity in Sera from BNT162b2 and mRNA-1273 Vaccine Recipients

A chaotrope ELISA was used to evaluate avidity over time (Figure 2). Vaccination with two doses of either mRNA vaccine induced avidity responses that significantly increased at each timepoint throughout 6 months (Figure 2A,B). No significant correlations were observed between age and avidity for either vaccine. Consistent with our previous report, which evaluated 1- and 4-month time points [13], the mRNA-1273 vaccine initially induced significantly higher anti-SARS-CoV-2 spike IgG avidity levels compared to the BNT161b2 vaccine following the primary vaccination series. The anti-SARS-CoV-2 spike IgG geometric mean avidity levels for mRNA-1273 were 3.1 M (95% CI, 3.0–3.3 M) at month 1, 3.4 M (95% CI, 3.2–3.6 M) at month 4 (month 1 to month 4, *p* = 0.011), and 3.9 M (95% CI, 3.7–4.0 M) at month 6 (month 4 to month 6, *p* ≤ 0.001), as shown in Figure 2A. The BNT162b2 recipients developed avidity levels of 2.5 M (95% CI, 2.4–2.7 M) at month 1, 3.0 M (95% CI, 2.8–3.2 M) at month 4 (month 1 to month 4, *p* ≤ 0.001), and 3.3 M (95% CI, 3.1–3.6 M) at month 6 (month 4 to month 6, *p* ≤ 0.001) (Figure 2B). The differences in avidity between the two vaccines were significant at all three timepoints (mRNA-1273 to BNT161b2: *p* ≤ 0.001, *p* = 0.002, and *p* = 0.002, respectively) (Figure 2C). These statistically higher avidity levels were achieved primarily in the male cohort (Figure 3A). The geometric mean avidity levels for the mRNA-1273 recipient male cohort were 3.1 M (95% CI, 2.9–3.3 M) at month 1, 3.4 M (95% CI, 3.2–3.6 M) at month 4, and 3.9 M (95% CI, 3.6–4.2 M) at 6 months. The male BNT162b2 recipients developed avidity levels of 2.5 M (95% CI, 2.3–2.7 M) at month 1, 2.6 M (95% CI, 2.4–2.9 M) at month 4, and 3.2 M (95% CI, 2.8–3.6 M) at month 6. The differences in avidity levels between the two vaccines for the males were significant at all three timepoints (mRNA-1273 to BNT162b2: *p* ≤ 0.001, *p* ≤ 0.001, *p* = 0.006, respectively). The females showed one instance of a statistically significant difference between the two vaccines, which was at month 1: mRNA-1273 3.1 M (95% CI, 2.9–3.4 M) versus. BNT162b2 2.6 M (95% CI, 2.4–2.8 M), *p* ≤ 0.001 (Figure 3B). While no differences were noted in the male versus female recipients of the mRNA-1273 vaccine (Figure 3C), it was noted that at one timepoint (month 4 following primary series vaccination), the female BNT162b2 recipients exhibited a statistically higher avidity when compared with the males (3.2 M versus 2.6 M, *p* = 0.008) (Figure 3D). 

### 3.5. Effects of a Homologous Third Dose on Anti-SARS-CoV-2 Spike IgG Avidity

The administration of a homologous third vaccine dose produced a substantial avidity increase in the first month, regardless of vaccine (*p* ≤ 0.001) (Figure 2A,B). No significant correlations were observed between sex or age and avidity levels after a third dose of either vaccine. While there was an immediate increase in IgG avidity for the SARS-CoV-2 spike after the third dose, there were few additional changes in avidity over the following 8 months (Figure 2A,B). The only significant change was observed in the mRNA-1273 cohort between 3 months and 6 months, which showed a slight decrease in the geometric mean avidity levels, which reached statistical significance (*p* = 0.006).

## 4. Discussion

Building on our prior investigations of COVID-19 mRNA vaccine immunogenicity in healthy individuals [13], we evaluated the magnitude of antibody level and avidity modulation up to 6 months after primary series vaccination with BNT162b2 or mRNA-1273 and up to 9 months after a third dose of the same vaccine (homologous booster vaccinations). Humoral protection through vaccination is typically mediated by neutralizing antibodies, and the concentrations of these effector molecules are generally accepted indicators of protection [23]. However, the specific level of circulating antibody that confers protection in humans against SARS-CoV-2 infection is not known, nor is the duration of immunity imparted by vaccination with the mRNA vaccines defined. In addition, antibody quality or avidity may play a role in protection against SARS-CoV-2 infection and disease. 

The role of avidity in protection against infection and disease is already well documented in multiple pathogens [24,25]. Serum antibodies with strong avidity responses have been shown to correlate with protection from malaria following vaccination with the RTS, S/ASO1E vaccine [24]. Meanwhile, serum antibodies with weak avidity responses following vaccination have been associated with vaccine failure, as with the *Haemophilus influenza* Type B vaccine [26], and are thought to be involved in disease enhancement of respiratory syncytial virus (RSV) and measles in vaccinated individuals [27,28]. In the context of COVID-19, the role of avidity is still under investigation [17,18,29]. 

Data are beginning to accumulate, suggesting that avidity may have clinical significance for SARS-CoV-2 immunity. Callaway et al. described an interesting role of avidity in SARS-CoV-2 variant immunity, demonstrating that high-avidity antibodies generated against the original strain of the virus were able to neutralize Omicron variants of SARS-CoV-2 in vitro [20]. These results are further supported by Harthaller et al., who found that individuals vaccinated against the original SARS-CoV-2 strain (with or without infection) with antibodies that neutralized the Omicron variant had higher avidity levels against the wild-type virus compared to those without Omicron-neutralizing antibodies [30]. 

Our data demonstrating the rapid decline in serum antibody levels are consistent with data from various studies reporting antibody waning over time [12,31,32]. The collective data and our study demonstrate the utility of a third vaccine dose, which reduces the rate of antibody level decline compared to primary series vaccination alone. Interestingly, while circulating anti-SARS-CoV-2 spike IgG declined rapidly in both vaccine groups over 6 months, avidity steadily increased. This increase is likely a sign of adaptive humoral immune response refinement via the selection of B cells most capable of binding antigens and with higher affinity. These B cells are expanded, and their resultant plasma cells secrete antibodies capable of higher binding strength, enhancing the net serum antibody avidity [21]. Through the administration of additional vaccine doses, the selection of the most capable B cells may be further expanded. 

Homologous third-dose vaccinations produced the expected anamnestic antibody responses to both vaccines, with IgG levels increasing in both cohorts. We noted a superior response to mRNA-1273 6 months post-dose 3, and this may be suggestive of a dose effect; a single dose of mRNA-1273 contains over three times the RNA content of a BNT162b2 dose [33], and mRNA vaccines may have self-adjuvating effects [34,35]. Anti-SARS-CoV-2 spike IgG levels then waned from 1 to 3 months after the third dose, but the rate of decline was substantially lower than what was detected after the primary series. These results are consistent with other studies [36,37]. 

As noted with the primary series vaccination, a third homologous vaccination dose significantly increased IgG avidity for the SARS-CoV-2 spike for both vaccines; however, avidity did not progressively increase over time as it did with the primary series doses. In this study, there was a leveling in antibody avidity that may be due to a limitation in the upper end of the assay. Regardless, these results highlight the utility of a third dose of mRNA vaccine to increase not only antibody levels but also antibody quality and, potentially, effectiveness. 

A limitation of this study is that it did not evaluate protection from illness, nor did it measure the levels of SARS-CoV-2-neutralizing antibodies in serum. In addition, nucleocapsid data are not reported. However, it has been previously reported that neutralizing antibodies correlate moderately to strongly with binding IgG levels [38,39,40,41]. In addition, Treibelhorn et al. found that vaccine nonresponders (<10 neutralizing units/mL following ≥ one vaccine dose) had significantly lower anti-SARS-CoV-2 spike IgG avidity levels compared to vaccine responders (>10 neutralizing units/mL following ≥ one vaccine dose) [42]. Future studies on the correlation between neutralizing antibody levels and antibody avidity in SARS-CoV-2 would be interesting in this cohort, but this was beyond the scope of the current study. However, since these results and the results of others indicate that avidity may be more stable than antibody levels, avidity may be a relevant marker of immunogenicity and protection against COVID-19 [43]. Further studies are needed to determine the relationship between avidity and neutralization and whether this increase in avidity is associated with protective immunity against multiple variants.

In addition to being a critical tool for evaluating clinical protection, serological data can also provide important information at the epidemiological level. Antibody avidity assays have been demonstrated to be useful for the identification of recent or multiple infections of various pathogens [44,45,46]. Golding et al. demonstrated that it is possible to measure SARS-CoV-2 reinfection via anti-SARS-CoV-2 nucleocapsid IgG avidity levels in nonlongitudinal serum samples [43]. Additional studies are needed for a better understanding of avidity. The number of breakthrough infections in this study was quite small and thus not investigated. A large prospective study would need to be performed to better understand the effects of breakthrough infections on avidity and the longevity of avidity levels over time. 

## 5. Conclusions

This study demonstrates that a third dose of either mRNA vaccine (mRNA-1273 or BNT162b2) elicits greater anti-SARS-CoV-2 spike IgG level durability and avidity compared with a primary two-dose regimen, suggesting that a third dose of mRNA vaccines provides more persistent protection from COVID-19. These data also indicate that recipients of mRNA-1273 initially elicit a stronger avidity response within 6 months after the primary vaccination compared to BNT162b2 recipients; however, following a three-dose regimen, there were no significant differences between the vaccines in terms of either the circulating IgG levels or avidity. Whether the newly available bivalent vaccines provide superior protection against circulating SARS-CoV-2 variants compared to the mRNA vaccines evaluated in this report is still under investigation. This study adds important avidity data to the growing knowledge landscape on antibody dynamics and kinetics in response to mRNA vaccination [47]. Although vaccine-elicited protection against SARS-CoV-2 infection wanes over time, it is worth highlighting that both of the FDA-authorized mRNA vaccines have been found to provide excellent protection against severe disease and death from SARS-CoV-2 and are critical tools for preventing death and hospitalization due to COVID-19 [48,49,50,51].

## Figures and Tables

**Figure 1 vaccines-12-00516-f001:**
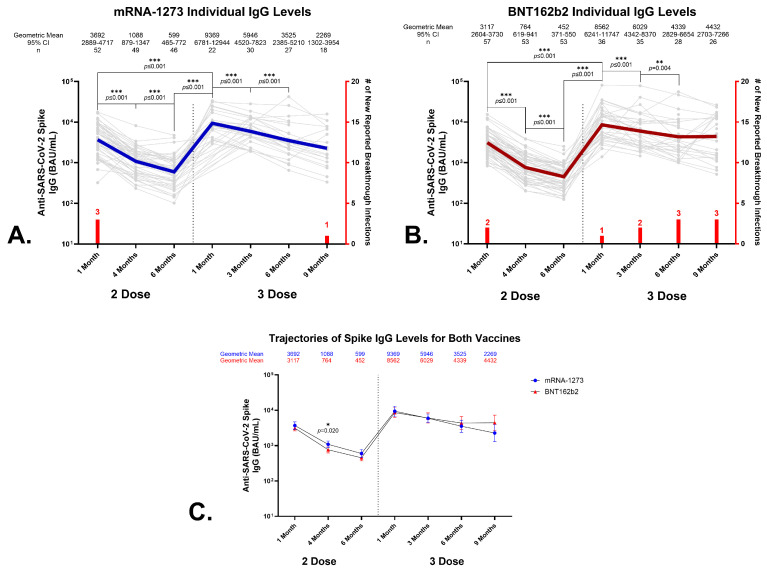
Anti-SARS-CoV-2 spike IgG levels in sera from vaccine recipients that received a homologous third dose of vaccine. Top panels: longitudinal analyses of anti-SARS-CoV-2 spike IgG levels of individual donors over 6 months post-primary series and 9 months following a homologous third dose of mRNA-1273 (**A**) or BNT162b2 (**B**). The geometric means, 95% confidence intervals (CIs), and number of recipients are listed above each timepoint. The bold, colored line (blue line, mRNA-1273; red line, BNT162b2) represents the geometric means, while the grey lines represent the individual recipients. The vaccine group geometric means and 95% confidence intervals (error bars) for the primary series and third dose for the mRNA-1273 and BNT162b2 recipients are shown in (**C**). Wilcoxon’s (Mann–Whitney test) *p* values are indicated, and *p* ≤ 0.05 *, *p* ≤ 0.01 **, and *p* ≤ 0.001 *** are considered significant. Red bars indicate the number of individuals self-reporting breakthrough infections at that timepoint.

**Figure 2 vaccines-12-00516-f002:**
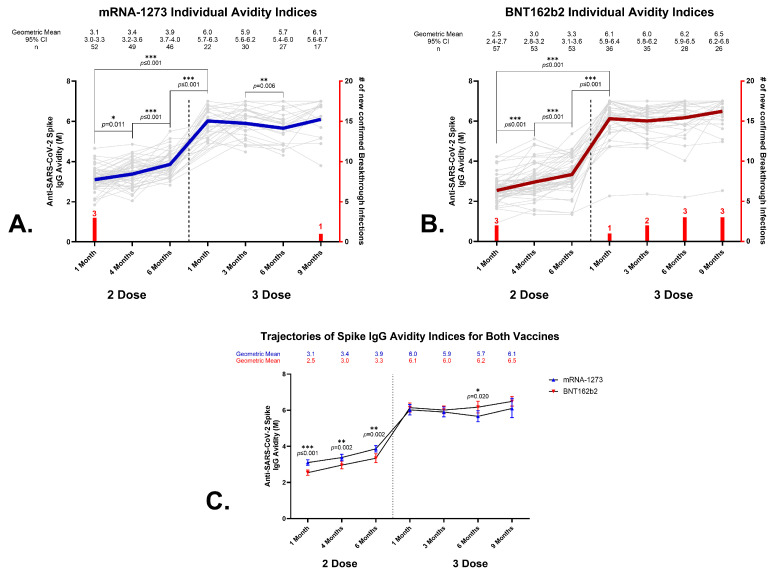
Anti-SARS-CoV-2 spike IgG avidity development in BNT162b2 and mRNA-1273 recipients that received a homologous third dose of vaccine. Top panels: longitudinal analyses of serum anti-SARS-CoV-2 spike IgG avidity (M) of individual recipients over 6 months who then received a homologous third dose with mRNA-1273 (**A**) or BNT162b2 (**B**). The geometric means, 95% confidence intervals (CIs), and number of recipients are listed above each timepoint. The bold, colored line (blue line, mRNA-1273; red line, BNT162b2) represents the geometric mean, while the grey lines represent the individual recipients. The vaccine group geometric means and 95% CI (error bars) for the primary series and the third dose after the primary series for the mRNA-1273 and BNT162b2 recipients are shown in (**C**). Significant *p* values calculated for each timepoint are indicated, and *p* ≤ 0.05 *, *p* ≤ 0.01 **, and *p* ≤ 0.001 *** are considered significant. Red bars indicate the number of individuals self-reporting breakthrough infections at that timepoint.

**Figure 3 vaccines-12-00516-f003:**
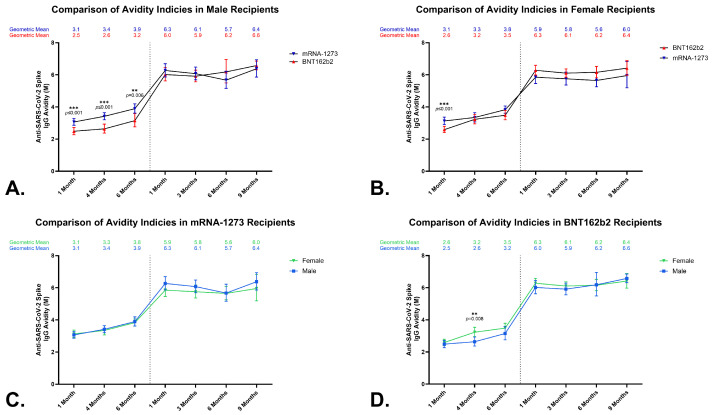
Comparison of anti-SARS-CoV-2 spike IgG avidity levels in BNT162b2 and mRNA-1273 recipients by sex. Top panels: longitudinal analyses of serum anti-SARS-CoV-2 spike IgG avidity (M) geometric means and 95% CIs (error bars) for primary series and third dose after primary series for mRNA-1273 and BNT162b2 recipients, separated by vaccine within a sex-at-birth group, males (**A**) or females (**B**). Bottom panels: longitudinal analyses of serum anti-SARS-CoV-2 spike IgG avidity (M) geometric means and 95% CIs (error bars) for primary series and third dose after primary series for mRNA-1273 and BNT162b2 recipients, separated by sex within a vaccine group, mRNA-1273 (**C**) or BNT162b2 (**D**). The geometric means are listed above each timepoint. Wilcoxon’s (Mann–Whitney test) *p* values are indicated, and *p* ≤ 0.01 **, and *p* ≤ 0.001 *** are considered significant.

**Table 1 vaccines-12-00516-t001:** Study demographics.

Participants	BNT162b2	mRNA-1273
Number (n)	57	52
Mean Age (Years)	47	44
Age Min–Max (Years)	23–66	23–67
Female (percent)	32 (56%)	28 (54%)
Male (percent)	25 (44%)	24 (46%)

**Table 2 vaccines-12-00516-t002:** Longitudinal assessment of participant demographics.

Participants	(A) Primary Series					
	BNT162b2			mRNA-1273		
	Number (n)	Time to Collection (Days ± SD)		Number (n)	Time to Collection (Days ± SD)	
Month 1	57	25 ± 11		52	20 ± 8	
Month 4	53	116 ± 12		49	114 ± 11	
Month 6	53	183 ± 18		46	192 ± 18	
Total receiving primary vaccine	57			52		

Participants	(B) Primary Series Plus Boost (third Dose) *					
	BNT162b2			mRNA-1273		
	Number (n)	Time to Collection (Days ± SD)		Number (n)	Time to Collection (Days ± SD)	
Month 1	36	37 ± 16		22	36 ± 18	
Month 3	35	95 ± 8		30	93 ± 7	
Month 6	28	183 ± 6		27	184 ± 6	
Month 9	26	278 ± 7		18	275 ± 5	
Total receiving primary vaccine	36			32		


* Not all participants contributed to every timepoint. SD, Standard Deviation.

## Data Availability

All data can be available upon request from the corresponding author.

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
