# Peer review of "Longitudinal Assessment of BNT162b2- and mRNA-1273-Induced Anti-SARS-CoV-2 Spike IgG Levels and Avidity Following Three Doses of Vaccination"

_vaccines, 2024, doi:10.3390/vaccines12050516_

Round 1

Reviewer 1 Report

Comments and Suggestions for Authors

The manuscript entitled "Longitudinal Assessment of BNT162b2- and mRNA-1273-induced anti-SARS-CoV-2 Spike IgG Levels and Avidity Following 3 Doses of Vaccination" shows that a third dose of SARS-CoV-2 mRNA vaccine boosts both the magnitude and the quality of specific IgG responses. While the circulating binding IgG level declines gradually after the third vaccination, the avidity remains quite stable. 

The study is well designed and I have no question about the data analysis and interpretation. However, I suggest to provide neutralizing antibody test data, despite neutralizing antibodies correlate with binding IgG levels. The binding IgG levels are similar at 1-month post the 2nd dose and 9-month post the 3rd month, while the avidity is obviously higher at the later time piont. It will be especially interesting to compare the neutralizing compacities at these two time pionts. 

Author Response

Thank you for this comment. Unfortunately, measuring antibody neutralization was beyond the scope of this study. We addressed this by adjusting the following to lines 335-349 in the discussion: “A limitation of this study is that it did not evaluate protection from illness, nor did it measure the levels of SARS-CoV-2 neutralizing antibodies in serum. However, it has been previously reported that neutralizing antibodies correlate moderately to strongly with binding IgG levels [40-43]. In addition, Treibelhorn et al. found that vaccine non-responders (< 10 neutralizing units/mL following ≥ 1 vaccine dose) had significantly lower anti- SARS-CoV-2 spike IgG avidity levels compared to vaccine responders (> 10 neutralizing units/mL following ≥ 1 vaccine dose) [44]. Future studies on the correlation between neutralizing antibody levels and antibody avidity in SARS-COV-2 would be interesting in this cohort but were beyond the scope of the current study. However, since these results and the results of others indicate avidity may be more stable than antibody levels, avidity may be a relevant marker for immunogenicity and protection against COVID-19 [45]. Further studies are needed to determine the relationship between avidity and neutralization, and whether this increase in avidity is associated with protective immunity against multiple variants.”

Reviewer 2 Report

Comments and Suggestions for Authors

This manuscript assesses the humoral immunity in COVID-19 mRNA vaccine recipients with long-term follow-up, including primary series and booster vaccination. The writing, structure, and findings were outstanding and interesting in this field. The overall quality of this manuscript is acceptable for publication.

Major concerns.

1. How to ensure that all participants did not get SARS-CoV-2 infection till to the study endpoint?

The immunologic outcomes (Figures 1A and 1B) show that some may get an infection during the follow-up.
The critical point is that hybrid immunity could alter the immunologic response in higher immunity levels and avidity (broader).

If all participants were not naïve to SARS-CoV-2 infection or this study was not screening for previous infection, such as anti-nucleocapsid.
Please add this issue to the limitation of this study.

2. What is the Institutional Review Board Statement of this study?

Minor concerns.

1. The mRNA-1273: The FDA recommendation for administering mRNA-1273 is 100 µg per dose for the primary series, and the booster vaccination is 50 µg per dose. However, some studies may use the same dose as primary series, especially in immunocompromised hosts.

Thus, it suggests clarifying the dose of each vaccine, both primary series and booster vaccination, in subsection 2.1.1 to prevent confusion in the future beyond the COVID-19 era.

Were the dosage of BNT162b2 30 µg, mRNA-1273 (primary) 100 µg, and mRNA-1273 (booster) 50 µg per dose?

Comments.

1. Line 32 and elsewhere: Suggests using "Pfizer— BioNTech" to make it consistent with the most common form. 

2. Suggest using "Moderna— NIAID", it is the same issue with Pfizer— BioNTech.

3. Was this study/project registered in any registry (e.g., ClinicalTrials.gov)?

If yes. You can add the registry's name with an identifier of this study/project to the statement in subsection 2.1.

4. This manuscript's scope on the immunoglobulin avidity. A further article on the antibody profile may benefit the Introduction and Discussion. (Optional)

For example;

Hosseinian S, de Assis R, Khalil G, Luu MK, Jain A, Horvath P, Nakajima R, Palma AM, Hoang A, Razzak E, Garcia N, Alger J, Kalantari M, Silzel EK, Jasinskas A, Zaldivar F, Schubl SD, Felgner PL, Khan S. Analysis and comparison of SARS-CoV-2 variant antibodies and neutralizing activity for 6 months after a booster mRNA vaccine in a healthcare worker population. Front Immunol. 2023 May 17;14:1166261. doi: 10.3389/fimmu.2023.1166261. PMID: 37266444; PMCID: PMC10229859.

Typos.

1. Line 30: "SARS-CoV-2".

2. Line 31: "coronavirus disease 2019 (COVID-19)".

3. Line 81: "-80°C"

4. Line 82: "56°C"

Author Response

Major concerns.

  1. How to ensure that all participants did not get SARS-CoV-2 infection till to the study endpoint?

The immunologic outcomes (Figures 1A and 1B) show that some may get an infection during the follow-up.
The critical point is that hybrid immunity could alter the immunologic response in higher immunity levels and avidity (broader).

If all participants were not naïve to SARS-CoV-2 infection or this study was not screening for previous infection, such as anti-nucleocapsid.
Please add this issue to the limitation of this study.

Thank you for this comment. We added the following to lines 67-68 “All participants self-reported as having no history of infection prior to the study” in the methods, and lines 337-338 to the discussion, reading “In addition, nucleocapsid data is not reported.” With the number of breakthrough infections included in Figure 1A, 1B, 2A, and 2B, the number of breakthrough infections in this study was quite small, and thus, not investigated in this study as highlighted in the discussion section lines 356-359.

  1. What is the Institutional Review Board Statement of this study?

The following was added to lines 63-69 of the methods section: “Sera was collected from 109 healthy donors by Occupational Health Services, at National Institutes of Health (NIH) National Cancer Institute (NCI) at Fort Detrick, MD, under the Research Donor Protocol (RDP). Participants were healthy NCI-Frederick employees and other NIH staff that donated blood samples for in vitro research at the NCI-Frederick laboratories. All participants self-reported as having no history of infection prior to the study. The protocol is listed under NIH protocol number OH99CN046 and NCT number NCT00339911.”

Minor concerns.

  1. The mRNA-1273: The FDA recommendation for administering mRNA-1273 is 100 µg per dose for the primary series, and the booster vaccination is 50 µg per dose. However, some studies may use the same dose as primary series, especially in immunocompromised hosts.

Thus, it suggests clarifying the dose of each vaccine, both primary series and booster vaccination, in subsection 2.1.1 to prevent confusion in the future beyond the COVID-19 era.

Were the dosage of BNT162b2 30 µg, mRNA-1273 (primary) 100 µg, and mRNA-1273 (booster) 50 µg per dose?

Thank you for this comment. The following was added to lines 69-75 to read “In this study, all donors had received 2 doses of either BNT162b2 (30 µg, n=57) or mRNA-1273 (100 µg, n=52). Samples were obtained at months 1, 4, and 6 following the 2 originally recommended doses (primary series). Administration of the 3rd dose of vaccine occurred 6 to 17 months after completion of the primary series, resulting in 68 vaccinated participants receiving a 3rd dose of the same vaccine as their primary (BNT162b2 30 µg, n=36; mRNA-1273 50 µg, n=32), and they were subsequently followed for up to 9 months.”

Comments.

  1. Line 32 and elsewhere: Suggests using "Pfizer— BioNTech" to make it consistent with the most common form. 

This was corrected in line 33 to read “BNT162b2 (Pfizer-BioNTech)”.

  1. Suggest using "Moderna— NIAID", it is the same issue with Pfizer— BioNTech.

Thank you for the comment. However, we have chosen to keep the name as “Moderna”, since this name is commonly used in the literature.

  1. Was this study/project registered in any registry (e.g., ClinicalTrials.gov)?

If yes. You can add the registry's name with an identifier of this study/project to the statement in subsection 2.1.

This study was not registered at ClinicalTrials.gov; however, the following was added to lines 63-69 of the methods section to highlight where the samples were collected from: “Sera was collected from 109 healthy donors by Occupational Health Services, at National Institutes of Health (NIH) National Cancer Institute (NCI) at Fort Detrick, MD, under the Research Donor Protocol (RDP). Participants were healthy NCI-Frederick employees and other NIH staff that donated blood samples for in vitro research at the NCI-Frederick laboratories. All participants self-reported as having no history of infection prior to the study. The protocol is listed under NIH protocol number OH99CN046 and NCT number NCT00339911.”

  1. This manuscript's scope on the immunoglobulin avidity. A further article on the antibody profile may benefit the Introduction and Discussion. (Optional)

A statement was added at lines 369-371 saying “This study adds important avidity data to the growing knowledge landscape on antibody dynamics and kinetics in response to mRNA vaccination (49).”

For example;

Hosseinian S, de Assis R, Khalil G, Luu MK, Jain A, Horvath P, Nakajima R, Palma AM, Hoang A, Razzak E, Garcia N, Alger J, Kalantari M, Silzel EK, Jasinskas A, Zaldivar F, Schubl SD, Felgner PL, Khan S. Analysis and comparison of SARS-CoV-2 variant antibodies and neutralizing activity for 6 months after a booster mRNA vaccine in a healthcare worker population. Front Immunol. 2023 May 17;14:1166261. doi: 10.3389/fimmu.2023.1166261. PMID: 37266444; PMCID: PMC10229859.

Typos.

  1. Line 30: "SARS-CoV-2". Corrected.
  2. Line 31: "coronavirus disease 2019 (COVID-19)". Corrected.
  3. Line 81: "-80°C" Corrected.
  4. Line 82: "56°C" Corrected.

Reviewer 3 Report

Comments and Suggestions for Authors

I was invited to revise the paper entitled "Longitudinal Assessment of BNT162b2- and mRNA-1273-induced anti-SARS-CoV-2 Spike IgG Levels and Avidity Following 3 Doses of Vaccination". It was a cohort study aimed to evaluate the quality and persistence of  immunity induced by mRNA vaccination over time and to evaluate the impact long-term avidity development. 

The topic is very interesting and few paper were published in my knowedge.

Observations:

- Generally, the word count should be improved, accordingly to journal guidelines;

- Sample size estimation was lacking;

- In table 1 and 2, study variables should be tested between groups;

- I'm a bit skeptikal about statistical analysis. In order to evaluate the impact of time, I suggest to perform a GLM or a Mixed model. A repeated Wilcoxon test, as performed by Authors, appear not appropriate;

- Authors should better clarify if patients were infected during the study or if someone was infected priorly the study start;

- I suggest also to perform a subanalysis by age and gender;

- Strenght and limitation section should be added.

Author Response

I was invited to revise the paper entitled "Longitudinal Assessment of BNT162b2- and mRNA-1273-induced anti-SARS-CoV-2 Spike IgG Levels and Avidity Following 3 Doses of Vaccination". It was a cohort study aimed to evaluate the quality and persistence of  immunity induced by mRNA vaccination over time and to evaluate the impact long-term avidity development. 

The topic is very interesting and few paper were published in my knowedge.

Observations:

- Generally, the word count should be improved, accordingly to journal guidelines;

The word count has been increased to 4506 words.

- Sample size estimation was lacking;

This is a research immune characteristics study utilizing a relatively small number of participants, so no power calculations were determined prior to the study. In addition, the lack of true avidity means and variances (due to the assay cut-offs) meant that power calculations could not be performed.

- In table 1 and 2, study variables should be tested between groups;

Study variables were tested between groups and were found to be insignificant except for a few cases addressed in the new figure 3. Statements clarifying this were added to lines 169-170, 184-186, 219-220, 231-245, and 249-251.

- I'm a bit skeptikal about statistical analysis. In order to evaluate the impact of time, I suggest to perform a GLM or a Mixed model. A repeated Wilcoxon test, as performed by Authors, appear not appropriate;

Avidity values in this assay had a maximum threshold of 7.0 M, so any sample with a potentially larger value was recorded as 7.0. The mean and variance of these values cannot be determined, so only nonparametric test could be used, precluding a GLM or mixed model analysis. A clarifying statement was added to lines 139-147.

- Authors should better clarify if patients were infected during the study or if someone was infected priorly the study start;

We added the following to lines 67-68 “All participants self-reported as having no history of infection prior to the study” in the methods, and lines 337-338 to the discussion, reading “In addition, nucleocapsid data is not reported.” With the number of breakthrough infections included in Figure 1A, 1B, 2A, and 2B, the number of breakthrough infections in this study was quite small, and thus, not investigated in this study as highlighted in the discussion section lines 356-359.

- I suggest also to perform a subanalysis by age and gender;

Study variables were tested between groups and were addressed through the new figure 3 and in lines 231-245, “These statistically higher avidity levels were achieved primarily in the male cohort (Figure 3A). Geometric mean avidity levels for the mRNA-1273 male cohort were 3.1 M (95% CI, 2.9-3.3 M) at month 1, 3.4 M (95% CI, 3.2-3.6 M) at month 4, and 3.9 M (95% CI, 3.6-4.2 M) at 6 months. Male BNT162b2 recipients developed avidity levels of 2.5 M (95% CI, 2.3-2.7 M) at month 1, 2.6 M (95% CI, 2.4-2.9 M) at month 4, and 3.2 M (95% CI, 2.8-3.6 M) at month 6. The avidity levels between the two vaccines for the males were significant at all three timepoints (mRNA-1273 to BNT162b2: p≤0.001, p≤0.001, p=0.006, respectively). The females showed one instance of a statistically significant difference between the two vaccines, which was at month 1: mRNA-1273 3.1 M (95% CI, 2.9-3.4 M) vs BNT162b2 2.6 M (95% CI, 2.4-2.8 M), p≤0.001 (Figure 3B). While no differences were noted in male versus female recipients of the mRNA-1273 vaccine (Figure 3C), it was noted that at one timepoint (month 4 following primary series vaccination) female BNT162b2 recipients exhibited a statistically higher avidity when compared with males (3.2 M vs 2.6 M, p=0.008) (Figure 3D).”

- Strenght and limitation section should be added.

Strengths and limitations have been addressed and more strongly emphasized in lines 336-359, “A limitation of this study is that it did not evaluate protection from illness, nor did it measure the levels of SARS-CoV-2 neutralizing antibodies in serum. In addition, nucleocapsid data is not reported. However, it has been previously reported that neutralizing antibodies correlate moderately to strongly with binding IgG levels [40-43]. In addition, Treibelhorn et al. found that vaccine non-responders (< 10 neutralizing units/mL following ≥ 1 vaccine dose) had significantly lower anti- SARS-CoV-2 spike IgG avidity levels compared to vaccine responders (> 10 neutralizing units/mL following ≥ 1 vaccine dose) [44]. Future studies on the correlation between neutralizing antibody levels and antibody avidity in SARS-CoV-2 would be interesting in this cohort but were beyond the scope of the current study. However, since these results and the results of others indicate avidity may be more stable than antibody levels, avidity may be a relevant marker for immunogenicity and protection against COVID-19 [45]. Further studies are needed to determine the relationship between avidity and neutralization, and whether this increase in avidity is associated with protective immunity against multiple variants.

In addition to being a critical tool for evaluation of clinical protection, serological data can also provide important information at the epidemiological level. Antibody avidity assays have been demonstrated to be useful for identification of recent or multiple infections in various pathogens [46-48]. Golding et al. demonstrated that it is possible to measure SARS-CoV-2 reinfection via anti-SARS-CoV-2 nucleocapsid IgG avidity levels in non-longitudinal serum samples [45]. Additional studies are needed for a better understanding of avidity. The number of breakthrough infections in this study was quite small, and thus, not investigated in this study. A large, prospective study would need to be performed to better understand the effects of breakthrough infection on avidity, and longevity of avidity levels over time.“